# Biosensors for *Klebsiella pneumoniae* with Molecularly Imprinted Polymer (MIP) Technique

**DOI:** 10.3390/s22124638

**Published:** 2022-06-20

**Authors:** Chuchart Pintavirooj, Naphatsawan Vongmanee, Wannisa Sukjee, Chak Sangma, Sarinporn Visitsattapongse

**Affiliations:** 1Department of Biomedical Engineering, School of Engineering, King Mongkut’s Institute of Technology Ladkrabang, Bangkok 10520, Thailand; chuchart.pi@kmitl.ac.th (C.P.); naphatsawan.v@hotmail.com (N.V.); 2Department of Chemistry, Faculty of Science, Kasetsart University, Bangkok 10900, Thailand; wannisa.sj@gmail.com (W.S.); fscicsm@ku.ac.th (C.S.)

**Keywords:** *Klebsiella pneumoniae*, nosocomial infection, molecular imprinting polymer

## Abstract

Nosocomial infection is one of the most important problems that occurs in hospitals, as it directly affects susceptible patients or patients with immune deficiency. *Klebsiella pneumoniae* (*K. pneumoniae)* is the most common cause of nosocomial infections in hospitals. *K. pneumoniae* can cause various diseases such as pneumonia, urinary tract infections, septicemias, and soft tissue infections, and it has also become highly resistant to antibiotics. The principal routes for the transmission of *K. pneumoniae* are via the gastrointestinal tract and the hands of hospital personnel via healthcare workers, patients, hospital equipment, and interventional procedures. These bacteria can spread rapidly in the hospital environment and tend to cause nosocomial outbreaks. In this research, we developed a MIP-based electrochemical biosensor to detect *K. pneumoniae*. Quantitative detection was performed using an electrochemical technique to measure the changes in electrical signals in different concentrations of *K. pneumoniae* ranging from 10 to 10^5^ CFU/mL. Our MIP-based *K. pneumoniae* sensor was found to achieve a high linear response, with an R^2^ value of 0.9919. A sensitivity test was also performed on bacteria with a similar structure to that of *K. pneumoniae*. The sensitivity results show that the MIP-based *K. pneumoniae* biosensor with a gold electrode was the most sensitive, with a 7.51 (% relative current/log concentration) when compared with the MIP sensor applied with *Pseudomonas aeruginosa* and *Enterococcus faecalis*, where the sensitivity was 2.634 and 2.226, respectively. Our sensor was also able to achieve a limit of detection (LOD) of 0.012 CFU/mL and limit of quantitation (LOQ) of 1.61 CFU/mL.

## 1. Introduction

At present, nosocomial infections or healthcare-associated infections are a significant issue and occur in patients under medical care in hospitals [1]. These infections frequently transpire in patients living in developed and developing countries [2] during hospital stays lasting 48–72 h or around 3 days, thus causing a prolonged stay in the hospital [3]. Prevalent infections include central line-associated bloodstream infections, catheter-associated urinary tract infections, surgical site infections, and ventilator-associated pneumonia [4]. Nosocomial pathogens include bacteria, viruses, and fungal parasites [5]. Nosocomial infections can affect the body in various ways and lead to many serious illnesses, and the premature mortality rate is frequently increasing and directly affects immunocompromised patients at high risk [1,3,4]. It has been reported that 1 out of 10 patients gets infected by nosocomial infection [6]. During hospitalization, these infections can be transmitted to patients by healthcare workers, other patients, hospital equipment, or interventional procedures [7].

*K. pneumoniae* is the most common and well-known nosocomial infection in hospital [8]. It is a Gram-negative bacterium with a rod shape of 2.0 by 0.5 µm in size on average [9]. *K. pneumoniae* infections include sepsis [10], pneumonia [11], urinary tract infections [12], bloodstream infections [13], wound or surgical site infections [14], meningitis [15], and hepatic abscess [16]. Moreover, the *K. pneumoniae* infections acquired in hospitals are difficult to treat [17] as many *K. pneumoniae* strains have acquired additional genetic traits and become highly resistant to treatment [18].

Recently, traditional laboratory techniques and biological methodologies have been used to identify *K. pneumoniae*, including microscopic determination [19], biochemical examination [20], and polymerase chain reaction (PCR), which has gained wide popularity for *K. pneumoniae* diagnostics [16]. However, these techniques are labor-intensive and time-consuming and also require expensive equipment and trained personnel. These technologies are best for diagnostics in the laboratory but are not suitable for on-site detection. Hence, rapid, accurate, and sensitive identification of *K. pneumoniae* is required for determining the most appropriate therapy and controlling the spread of the pathogen [21]. Nowadays, most research about *K. pneumoniae* detection is based on foodborne pathogens and uses complex methods for analysis. 

In this work, we used electrochemical biosensors to analyze the detection result. Electrochemical sensors are one of the most common sensor types [22,23]. This type of sensor works with a combination of three electrodes, namely a working electrode, a counter electrode, and a reference electrode. A signal occurs when the chemical interaction changes on the surface of the electrode. This sensor has crucial advantages such as its size, portability, economic features, and ability to be used as a point-of-care device by patients themselves at home or in a doctor’s clinic, which also makes the electrochemical sensors suitable for sensing applications [24,25].

The molecular imprinting polymer technique is a molecular recognition using polymer synthesis to be an artificial receptor compatible with a specific molecule. MIPs are designed to mimic biological recognition such as enzyme and substrate, antigen, and antibody. This method has been widely used in various applications such as pharmaceutics [26], the food industry [27], drug delivery systems [26], and biosensors. MIP-based electrochemical sensors are currently being used for detection, and it is cost-effective to design sensors that can recognize biomolecules [28] such as bacteria [29,30], viruses [29,31], proteins [32], enzymes [33], amino acids [34], antibodies [35], and drugs [36]. However, the use of larger and more complex structures for the imprinting of bacterial cells challenges some characteristics and natural structures of microorganisms [37]. The MIP method is principally based on the specific molecular recognition of a template molecule, and a MIP can be synthesized using different types and combinations of functional monomers, cross-linkers, initiators, and solvents in the optimum ratio [28]. Due to its multiple advantages, including its easy synthesis and cost-effectiveness for preparation, and high stability, high affinity, and high selectivity to the template molecule [28,38,39,40], MIP-based electrochemical sensors have drawn researcher attention recently. Sharma et al. [41] described the synthesis of MIP and electrochemical sensing of *K. pneumoniae*. Pyrrole was used as monomer to imprint *K. pneumoniae,* which was later removed ultrasonication. The developed MIP-based sensor of Sharma et al. achieved a sensitivity of 3 μA mL CFU/cm^2^, LOD 1.352 CFU/mL, and linear detection range of 1 to 10^5^ CFU per mL. Sukjee et al. [42] presented a MIP sensor for measuring SARS-CoV-2 whole-virus particles in the environment. Four monomers with the optimized ratio of acrylamide (AAM), methacrylic acid (MAA), methyl methacrylate (MMA), and N-vinylpyrrolidone (NVP) were used in the study. This sensor can detect SARS-CoV-2 at concentrations as low as 0.1 fM in a buffer and samples prepared from reservoir water with a three log-scale linearity. Türkmen et al. [43] applied molecular imprinting polymer technique with surface plasmon resonance to detect *Pseudomonas* sp. bacteria in real-time. The detection of limit value was calculated as 0.5 × 10^2^ CFU/mL. This sensor system provided inexpensive, easy, fast, high sensitivity, and selectivity with excellent potential for bacterial analysis in food samples. A more in-depth review of MIP-based electrochemical sensors can be referred to in previous works [44,45,46,47].

In this work, we studied the use of electrochemical biosensors for *K. pneumoniae* detection using the MIP technique. The analysis in this work assesses electrochemical biosensor-based cyclic voltammetry (CV) detection methods depending on the observation of current changes happening at the sensor surface due to the biochemical interactions on the electrode [48]. This technique has many features that are suitable for *K. pneumoniae* detection and offers rapid response, selectivity, high sensitivity, and ease of use. The composites of functionalized material for *K. pneumoniae* detection such as the imprinting process and conductive nanoparticles and the specificity for binding *K. pneumoniae* are also determined in this work. Our work is in fact similar to the research work of Sharma et al. [41]. What differentiate our works from Sharma et al. study, however, is that our study explores on different types of monomers and the optimization of the monomer ratio used in our MIP-based electrochemical biosensors which make our sensor more suitable and reliable for *K. pneumoniae* detection.

## 2. Materials and Methods

### 2.1. K. pneumoniae Culturing and Growth Conditions

*K. pneumoniae* (TISTR 1383) was prepared in a biosafety cabinet and streaked on a Luria–Bertani (LB) agar plate before being incubated at 37 °C for 16 h. A single colony was isolated and inoculated into the LB broth and incubated at 240 rpm at 37 °C for 16 h. The *K. pneumoniae* culture was serially diluted to quantify the bacterial colony-forming units (CFU/mL). The value of the original stocks was determined using the viable count spread plate method. This method can produce stocks with approximately 4.2 × 10^9^ CFU/mL. For template imprinting, 1 µL *K. pneumoniae* stock with approximately 4.2 × 10^6^ CFU was used for coating on the working electrode. Dilution was performed in a liquid suspension of phosphate-buffered saline (PBS) at pH 7.4 to a concentration range of 10^1^–10^5^ CFU/mL as the analytical range for this work as shown in Figure 1.

### 2.2. K. pneumoniae Fixation for Imprinting on the Electrode and Scanning Electron Microscope (SEM) Sample Preparation

First, the bacteria grown in LB broth were washed with PBS to remove mucus or any other contaminant before imprinting on the working electrode [49]. To remove mucus, the electrode was washed with PBS 5 times. The electrode was fixed with 1% glutaraldehyde and then processed for 60 min, followed by 3% glutaraldehyde overnight. The electrode was then dehydrated with sequential concentrations of ethanol of 30%, 50%, 70%, 80%, 90%, 95%, and 100%. Each step lasted for 15 min, except 100% which lasted for 30 min.

A Quanta 450 FEI instrument was used for the SEM analysis to examine the morphology of the *K. pneumoniae*. Prior to SEM analysis, the bacterial samples were fixed on glass slides, dehydrated with the ethanol series mentioned above, and then dried in a critical point dryer machine to replace water with carbon dioxide (CO_2_) in the cells. After that, the samples were coated with 40–60 nm of gold material to obtain electrolytic conductance, resulting in high-quality SEM visualization.

### 2.3. Estimating Number of Imprinted K. pneumoniae Cells

The number of imprinted *K. pneumoniae* cells has been show to affect the performance of the MIP sensor. Ideally, the number of imprinted *K. pneumoniae* cells can be estimated from the working electrode area/*K. pneumoniae* cell area ratio. In practice, however, due to the randomly arranged distribution of *K. pneumoniae* in the solution, the number of imprinted *K. pneumoniae* cells in our study was adjusted by 10-fold dilution. The area of our designed working electrode on SPE was 12.56 mm^2^, and the area of *K. pneumoniae* was approximately 8.5 × 10^−7^ mm^2^/cell. In calculation, 1.48 × 10^7^
*K. pneumoniae* cells were imprinted on the working electrode. In practice, however, 10^6^ cells are normally needed for the imprinting process to prevent overlap of the imprinted bacteria on working electrode.

### 2.4. Polymers Synthesis for Bacterial Sample Detection

Dynamic light scattering (DLS) was used to investigate the type of charge around the surface of *K. pneumoniae* by adding a *K. pneumoniae* culture into a DLS chamber for potential measurement. The result of the DLS procedure showed that the voltage around the surface of *K. pneumoniae* was approximately −10 mV. The polymer synthesized for *K. pneumoniae* detection was made from positive monomers, as the surface of *K. pneumoniae* was negatively charged. The optimal ratio of the functional monomer was needed to ensure a high binding capacity for the bacterial cell and cross-linking agent for material hardening. Functional monomers were bound with *K. pneumoniae* to form a pre-polymerization complex. The formation of this complex is essential for detecting *K. pneumoniae* in a sample. MAM and AAM were chosen as positive monomers. NVP was chosen as a hydrophobic monomer. A hydrophobic group is better than a hydrophilic group because when bacteria imprint on a prepolymer complex, the hydrophilic group will destroy the structure and shape of the template. Azobis(isobutyronitrile) (AIBN) was selected as an initiator for complete polymerization. Additionally, N,N′-(1,2-dihydroxyethylene) bisacrylamide (DHEBA) was chosen as a cross-linker for template hardening on the prepolymer complex and Dimethyl sulfoxide (DMSO) was used as a solvent.

### 2.5. Fabrication of Screen-Printed Electrode (SPE)

We investigated two different types of MIP sensor electrodes, including a screen-printed electrode coated with carbon and one with gold and reduced graphene oxide (GO). GO has lower conductivity than graphene, so reduced GO is more often employed as an electrode modifier in the electrochemical biosensing area [50]. The steps followed for *K. pneumoniae* template imprinting are shown in Figure 2. The screen-printed electrode (SPE) in this work had gold as the working and counter electrode material and silver as the reference electrode material (DRP-220BT, Dropsens, Asturias, Spain). The sensitive layer was coated on the working SPE (4-mm diameter). This biosensor was made from polymer and graphene oxide. Table 1 shows the ratio in composition of the monomer mixture with two positive monomers (MAM and AAM) and one hydrophobic polymer (NVP). Each polymer condition was mixed with 1.5 mg of the free radical initiator AIBN and 47 mg of the cross-linker DHEBA. Further, 300 μL of DMSO was mixed in. The mixture was pre-polymerized at 70 °C to obtain a gel solution. Then, the prepolymer gel was mixed with 0.15 mg/mL GO at a ratio of 2:3 before coating on the WE. Then, 1 µL of prepolymer was mixed with GO coated on the working electrode (WE). Additionally, 1 µL of *K. pneumoniae* template was dropped on the polymer mixed with GO for imprinting on the SPE, followed by exposure to UV light for 3 h and incubation at 55 °C for 15 h to complete the template self-assembly and polymerization process. After that, the bacterial template was removed by washing with 10% acetic acid for 30 min followed by distilled water at 50 °C for 30 min [31]. All procedures were conducted in a biosafety cabinet.

### 2.6. Electrochemical Characterization

The cyclic voltammetry (CV) experiment was conducted using AutoLab PGSTAT302N (Methrom Dropsens, Asturias, Spain). *K. pneumoniae* samples were prepared by diluting the stock to the desired concentrations of 10^1^–10^5^ CFU/mL in 0.01 M phosphate-buffered saline (PBS) containing 5 mM K_4_Fe(CN)_6_/K_3_Fe(CN)_6_ in a ratio of 1:1 as a redox couple. The CV potential was scanned from −0.3 to +0.6 V at a scan rate of 50 mV/s. The process started with adding 10 μL of *K. pneumoniae* analyte on the prepolymer gel with GO on a gold electrode followed by measuring the current change with a potentiostat and drying on the electrode before the next CV measurement at any concentration.

## 3. Results

### 3.1. SEM Images of K. pneumoniae MIPs on the SPE Electrode

To confirm the successful imprinting of *K. pneumoniae* on the MIP sensor, an electron microscope was used. Figure 3a shows the rod-shaped morphology of *K. pneumoniae* in an SEM image. Figure 3b shows the cavity of the imprinted *K. pneumoniae* cell template with a size of approximately 0.5 × 1.7 µm on the working electrode after the imprinting process. The zeta potential of *K. pneumoniae* showed an average of approximately −10.6 mV. This result showed that a negative charge existed around the surface of *K. pneumoniae*; thus, we chose positive monomers to prepare the polymer for *K. pneumoniae* detection.

### 3.2. Polymers Synthesis and Selectivity for K. pneumoniae Detection

As mentioned above, the polymer synthesized for *K. pneumoniae* detection was made from positive monomers because there is a negative charge around the surface of *K. pneumoniae*. The resulting bacterial surface charges are shown in Table 2 Functional positively charged monomers must interact with the target molecule in order to form a pre-polymerization complex, which is essential for the formation of specific binding sites to the template. In this work, we chose MAM and AAM as positive monomers and NVP as a hydrophobic monomer. A hydrophobic group is better than a hydrophilic group because when bacteria imprint on a prepolymer, this type cannot destroy the structure and shape of the template. AIBN was selected as a radio initiator for complete polymerization. DHEBA was selected as a cross-linker for template hardening on the prepolymer. DMSO was used as a solvent, as it is a colorless liquid and a powerful solvent that dissolves both polar and non-polar compounds and is used in a wide range of organic solvents, as well as water. Furthermore, it can easily dissolve many substances that have extremely poor solubility in water.

For selectivity testing, *Pseudomonas aeruginosa* (*P. aeruginosa*) bacteria, which are of similar shape, size, and surface charge to *K. pneumoniae*, were chosen. To compare the signals that occur, Gram-positive *Enterococcus faecalis* (*E. faecalis*) bacteria, which cause nosocomial infections in hospitals, were also selected for selectivity testing. The surface charges of these bacteria are shown in Table 2.

### 3.3. Cyclic Voltammogram in Each Condition of Monomers on Carbon Electrode

As the optimal ratio of monomers is crucial for the effective performance of a *K. pneumoniae* MIP sensor, 10 rough ratios of MAM, AAM, and NVP, named conditions 1 to 10, were investigated. The ratios of the mixed monomers MAM:AAM:NVP in conditions 1 to 10 were 2:3:1, 4:2:1, 2:5:1, 2:7:1, 2:2:1, 1:1:1, 1:2:1, 1:4:1, 2:1:1, and 4:1:1, respectively. From the cyclic voltammograms, the blank electrode current and peak current of each concentration for each condition were measured and are shown in Table 3. To compare the performance of the electrodes, percent relative current was used, which is defined as follows:(1)$$\%\;Relative\;Current=\frac{\left(I-I_{0}\right)}{I_{0}}\times 100$$
where *I* is the current for each MIP-based sensor of various concentrations of *K. pneumoniae* and *I*_0_ is the current of the sensor when without of the *K. pneumoniae* concentration or the sensor is blank. A characteristic graph of % relative current versus log concentration was plotted. A comparison of the linearity condition of the monomers in all 10 polymer conditions on the carbon electrode is shown in Figure 4.

This characteristic graph of % relative current versus log of concentration shows that all conditions did not give a linear response, especially at low concentrations of *K. pneumoniae*. Specifically, the graphs of conditions 1, 2, 3, 4, 6, 7, 8, and 10 (ratios of 2:3:1, 4:2:1, 2:5:1, 2:7:1, 1:1:1, 1:2:1, 1:4:1, and 4:1:1 respectively) show a negative response at low concentration, and conditions 5 and 9 demonstrated a positive response. In conditions 5 and 9 (MAM:AAM:NVP ratio of 2:2:1 and 2:1:1), the best results were based on its positive response in the characteristic graph at a higher concentration level. However, conditions 5 and 9 performed poorly for *K. pneumoniae* detection at a low concentration. Hence, conditions 5 and 9 were selected for further investigation.

### 3.4. Study on Type of Electrode: Gold and Carbon Electrode

The type of electrode used was further investigated to find the optimal MIP sensor for *K. pneumoniae* detection. Conditions 5 and 9 were selected to test the gold electrode, as shown in Figure 5a and Figure 5b, respectively. In Table 4 and Figure 6, the data shows a characteristic graph of % relative current versus log of concentration on the gold electrode. Based on the graph, it is clear that condition 9 provides an optimal condition for the MIP sensor, as it has the highest linearity and specificity. Condition 5 also proved that it could provide a comparable and acceptable performance. When considering the magnitude of current that will result in a better dynamic range, condition 9 proved to be superior to condition 5 and, hence, was selected to be the optimal MIP sensor for *K. pneumoniae* detection.

### 3.5. Specificity Test

The cyclic voltammograms from the carbon electrode and gold electrode experiments showed the ability to differentiate the signals from different *K. pneumoniae* concentrations and were used to confirm the successful imprinting. The decrease in current gain changed with the increasing *K. pneumoniae* concentration in the analyte sample on the electrode, indicating that the absorption of *K. pneumoniae* on the surface of the working electrode biosensor increased the impedance. The specificity of *K. pneumoniae* was validated by comparing with the signals of current for *P. aeruginosa* and *E. faecalis* as a cross-selectivity measurement. Both types of bacteria are found in almost the same environment, and they have a similar morphology to *K. pneumoniae*. At high concentrations of *P. aeruginosa* and *E. faecalis*, the cyclic voltammograms showed that the current did not change by more than 5%, whereas the result of *K. pneumoniae* at each concentration showed the current changing by more than 5%. Figure 7 shows that the linearity observed at up to four log-scale concentrations ranged from 10^1^ to 10^5^ CFU/mL. In practice, the LOD for *K. pneumoniae* was found to be approximately 10 CFU/mL. In the reusable electrode test, the *K. pneumoniae* SPE could be used two times with the same electrode, but the second time, the performance was reduced by 15%. In calculation, the LOD was calculated from 3.3 × (σ/slope) and the LOQ was defined as 10 × (σ/slope). Our sensor was able to achieve an LOD of 0.012 CFU/mL and LOQ of 1.61 CFU/mL.

## 4. Discussion

This research was focused on the design and fabrication of an MIP-based sensor for the detection of *K. pneumoniae*, a common cause of nosocomial infection. The results demonstrate that the sensor can be used to detect *K. pneumoniae* with high sensitivity, linearity, and specificity. There are, however, a number of issues that need to be discussed for completeness:(i).In the research, various ratios of three functional monomers, constituting the polymer—namely MAM, AAM, and NVP—were evaluated for optimization, and the results demonstrated that the combination of the three monomers at a ratio of 2:1:1 (condition 9) gave the best linearity performance in *K. pneumoniae* detection.(ii).Testing on the gold electrode provides an effective performance for MIP sensor for *K. pneumoniae* with high sensitivity calculated by slope of linearity equation of 7.51 (% relative current/log concentration). For the reproducibility of this sensor, it can give approximately 5% of an error bar in the linearity range of log concentration levels.(iii).To demonstrate the specificity of our MIP-based *K. pneumoniae* sensor, we tested it with *E. faecalis* and *P. aeruginosa*, which are found in most environments. The sensitivity of the MIP-based *K. pneumoniae* biosensor with the gold electrode was the highest at a value 7.51 (% relative current/log concentration) compared with the MIP sensor applied with *P. aeruginosa* and *E. faecalis*. The sensitivity values of the MIP sensor applied with *P. aeruginosa* and *E. faecalis* were 2.634 and 2.226 (% relative current/log concentration), respectively.(iv).The experimental results show that the MIP-based *K. pneumoniae* sensor provides not only a linear response but also high specificity. Our optimal choice for the best linearity was obtained with an MAM:AAM:NVP ratio of 2:1:1 in the gold electrode MIP sensor with an R^2^ value of 0.9919 and best sensitivity of 7.51 (% relative current/log concentration). The LOD for *K. pneumoniae* was found to be approximately 10 CFU/mL in a practice experiment and was found to be 0.012 CFU/mL by calculation, which was superior to [41] where the LOD was reported to be 1.352 CFU/mL. For the LOQ of our MIP sensor, it achieved 1.61 CFU/mL.

## 5. Conclusions

This work demonstrated a method to develop biosensors based on MIP specifically for the rapid detection of *K. pneumoniae* and using electrochemical biosensors for analysis. Various conditions have been optimized for our designed MIP-based sensor, including the ratio of monomers and the type of electrode. The optimal ratio of MAM:AAM:NVP to detect *K. pneumoniae* was found to be 2:1:1. Both the carbon electrode and the gold electrode, used with a 2:1:1 ratio for the polymer, can achieve an acceptable performance. The sensitivity of the gold electrode was 7.51 (% relative current/log concentration). The best linearity response was also found in the gold electrode with an R2 value of 0.9919. Our MIP-based sensor also had a detection limit of 10 CFU/mL, which was found to be sufficient for detecting *K. pneumoniae* in real-world applications. Specificity testing of the MIP-based sensor on the rod-type bacteria was also conducted, which confirmed that the MIP demonstrates a lower sensitivity response in other rod-type bacteria. The MIP-based *K pneumoniae* sensor is easy to analyze, provides fast detection for diagnostics, and has a high accuracy and specificity for *K. pneumoniae* detection.

## Figures and Tables

**Figure 1 sensors-22-04638-f001:**
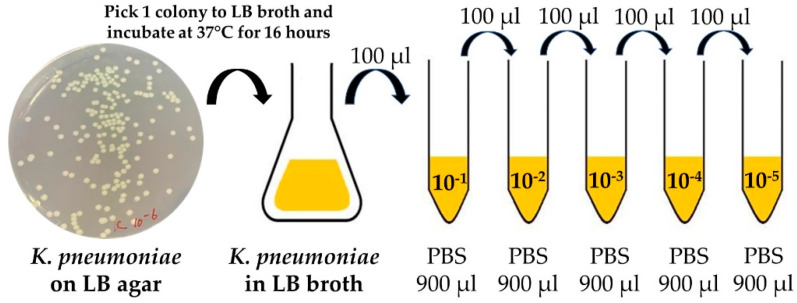
Schematic representation of *K. pneumoniae* on LB agar and the serial dilution method.

**Figure 2 sensors-22-04638-f002:**
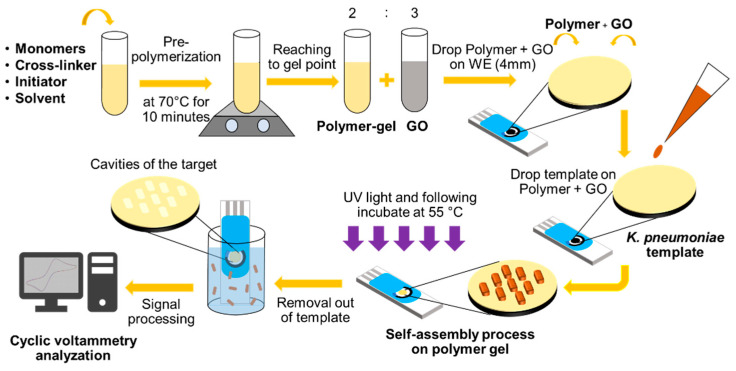
Schematic representation of the preparation of polymer-GO on gold electrode for *K. pneumoniae* detection.

**Figure 3 sensors-22-04638-f003:**
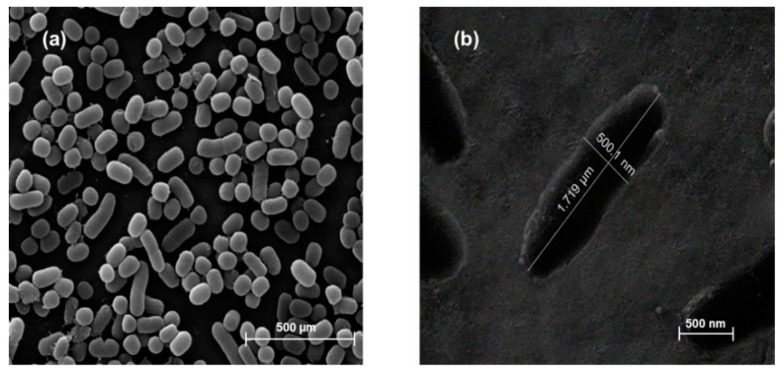
(**a**) Magnified SEM image at 10,000×; (**b**) the surface of whole *K. pneumoniae* on the SPE, with a size of approximately 0.5 × 1.7 µm (SEM image at 50,000×).

**Figure 4 sensors-22-04638-f004:**
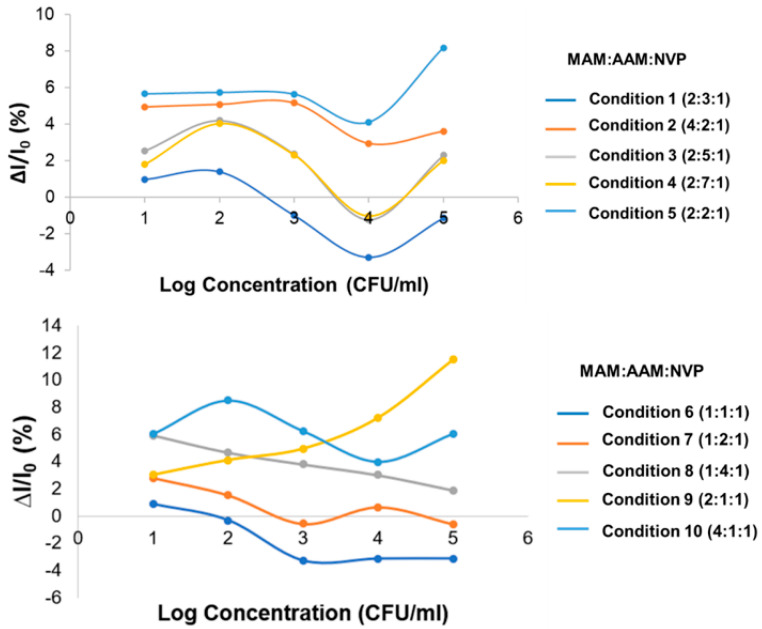
Comparison of linearity range for conditions 1 to 10 on the carbon electrode.

**Figure 5 sensors-22-04638-f005:**
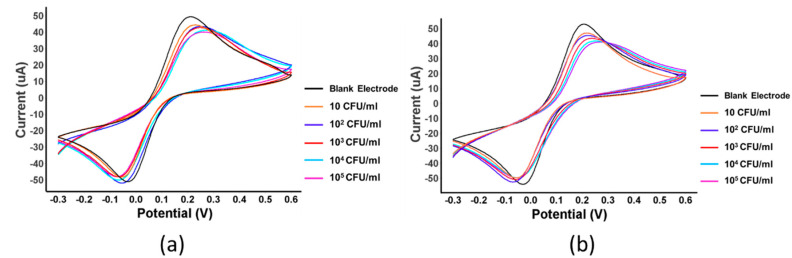
(**a**) Cyclic voltammogram of *K. pneumoniae* at concentration levels in condition 5 on gold electrode; (**b**) cyclic voltammogram of *K. pneumoniae* at concentration levels in condition 9 on gold electrode.

**Figure 6 sensors-22-04638-f006:**
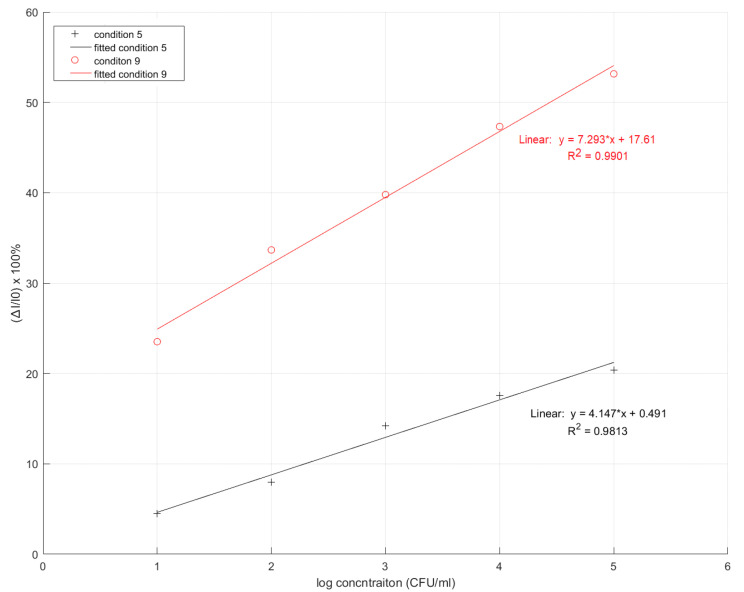
Comparison of linearity range for condition 5 and condition 9 on the gold electrode (* is referred to multiplication).

**Figure 7 sensors-22-04638-f007:**
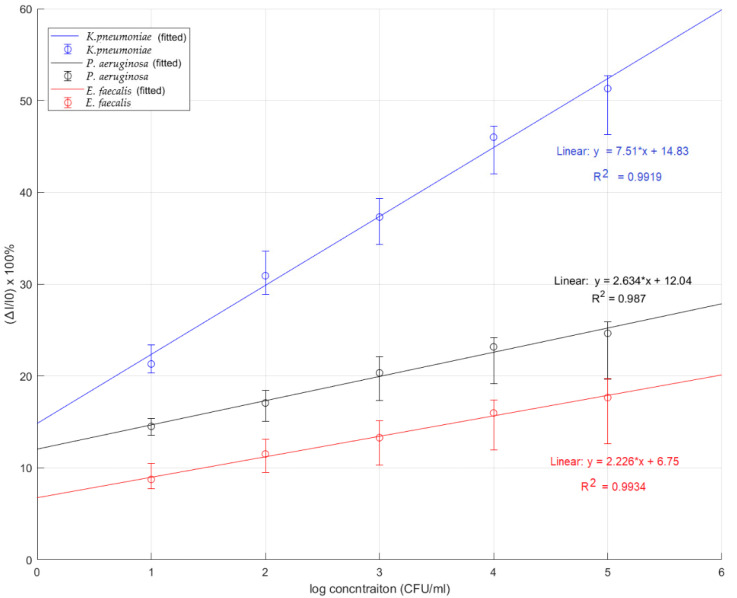
The plot of current changes as a linear function of log *K. pneumoniae* concentration gains from MIP composite on the gold electrode compared with the results from the control experiments using *P. aeruginosa* and *E. faecalis.* The number of replicated measurements is 3. (* is referred to multiplication).

**Table 1 sensors-22-04638-t001:** The amount of each functional monomer in various conditions *.

Condition	Ratio (n:n:n)	Methacrylamide (MAM) (mg)	Acrylamide (AAM) (mg)	N-Vinylpyrrolidone (NVP) (µL)
**1**	2:3:1	17	21.3	10.7
**2**	4:2:1	34	14.2	10.7
**3**	2:5:1	17	35.5	10.7
**4**	2:7:1	17	49.7	10.7
**5**	2:2:1	17	14.2	10.7
**6**	1:1:1	8.5	7.1	10.7
**7**	1:2:1	8.5	14.2	10.7
**8**	1:4:1	8.5	28.4	10.7
**9**	2:1:1	17	7.1	10.7
**10**	4:1:1	34	7.1	10.7

* 47 mg of N,N′-(1,2-dihydroxyethylene)bisacrylamide (DHEBA) has been used as crosslinker.

**Table 2 sensors-22-04638-t002:** The zeta potential around the surface of bacterial samples from dynamic light scattering.

Record	Bacteria Sample	Average Zeta Potential (mV)
1	*K. pneumoniae*	−10.6
2	*P. aeruginosa*	−10.1
3	*E. faecalis*	−16.2

**Table 3 sensors-22-04638-t003:** The current data of conditions 1–10 on the carbon electrode.

Condition	Concentration	Current (µA)	∆I (µA)	(∆I/I_0_) × 100%
Condition 1 (MAM:AAM:NVP) (2:3:1)	Blank	52.24	-	-
	10 CFU/mL	51.73	0.51	0.97
	10^2^ CFU/mL	51.51	0.73	1.40
	10^3^ CFU/mL	52.75	−0.52	−0.99
	10^4^ CFU/mL	53.95	−1.71	−3.27
	10^5^ CFU/mL	52.83	−0.59	−1.13
Condition 2 (MAM:AAM:NVP) (4:2:1)	Blank	45.56	-	-
	10 CFU/mL	43.31	2.25	4.94
	10^2^ CFU/mL	43.25	2.31	5.08
	10^3^ CFU/mL	43.22	2.35	5.15
	10^4^ CFU/mL	44.22	1.34	2.95
	10^5^ CFU/mL	43.92	1.65	3.61
Condition 3 (MAM:AAM:NVP) (2:5:1)	Blank	54.81	-	-
	10 CFU/mL	53.42	1.39	2.54
	10^2^ CFU/mL	52.51	2.30	4.20
	10^3^ CFU/mL	53.52	1.29	2.36
	10^4^ CFU/mL	55.49	−0.67	−1.23
	10^5^ CFU/mL	53.56	1.26	2.29
Condition 4 (MAM:AAM:NVP) (2:7:1)	Blank	53.87	-	-
	10 CFU/mL	52.90	0.97	1.80
	10^2^ CFU/mL	51.70	2.17	4.03
	10^3^ CFU/mL	52.62	1.25	2.32
	10^4^ CFU/mL	54.43	−0.56	−1.03
	10^5^ CFU/mL	52.79	1.09	2.01
Condition 5 (MAM:AAM:NVP) (2:2:1)	Blank	19.04	-	-
	10 CFU/mL	17.96	1.08	5.66
	10^2^ CFU/mL	17.94	1.09	5.74
	10^3^ CFU/mL	17.96	1.08	5.65
	10^4^ CFU/mL	18.25	0.78	4.12
	10^5^ CFU/mL	17.48	1.56	8.17
Condition 6 (MAM:AAM:NVP) (1:1:1)	Blank	49.06	-	-
	10 CFU/mL	48.63	0.42	0.87
	10^2^ CFU/mL	49.22	−0.16	−0.33
	10^3^ CFU/mL	50.67	−1.62	−3.30
	10^4^ CFU/mL	50.60	−1.54	−3.15
	10^5^ CFU/mL	50.60	−1.54	−3.14
Condition 7 (MAM:AAM:NVP) (1:2:1)	Blank	46.26	-	-
	10 CFU/mL	44.97	1.29	2.79
	10^2^ CFU/mL	45.56	0.70	1.52
	10^3^ CFU/mL	46.52	−0.26	−0.57
	10^4^ CFU/mL	45.96	0.30	0.65
	10^5^ CFU/mL	46.54	−0.27	−0.59
Condition 8 (MAM:AAM:NVP) (1:4:1)	Blank	49.94	-	-
	10 CFU/mL	46.98	2.96	5.92
	10^2^ CFU/mL	47.62	2.33	4.66
	10^3^ CFU/mL	48.05	1.90	3.80
	10^4^ CFU/mL	48.44	1.50	3.01
	10^5^ CFU/mL	48.99	0.95	1.90
Condition 9 (MAM:AAM:NVP) (2:1:1)	Blank	52.25	-	-
	10 CFU/mL	50.67	1.58	3.02
	10^2^ CFU/mL	50.10	2.15	4.11
	10^3^ CFU/mL	49.67	2.58	4.94
	10^4^ CFU/mL	48.48	3.77	7.22
	10^5^ CFU/mL	46.24	6.01	11.50
Condition 10 (MAM:AAM:NVP) (4:1:1)	Blank	49.88	-	-
	10 CFU/mL	46.87	3.01	6.03
	10^2^ CFU/mL	45.64	4.24	8.50
	10^3^ CFU/mL	46.77	3.11	6.23
	10^4^ CFU/mL	47.90	1.98	3.97
	10^5^ CFU/mL	46.86	3.02	6.05

**Table 4 sensors-22-04638-t004:** The current data of conditions 5 and 9 on gold electrode.

Condition	Concentration	Current (µA)	∆I (µA)	(∆I/I_0_) × 100%
Condition 5 (MAM:AAM:NVP) (2:2:1)	Blank	65.14	-	-
	10 CFU/mL	62.23	2.91	4.47
	10^2^ CFU/mL	59.95	5.20	7.98
	10^3^ CFU/mL	55.88	9.26	14.22
	10^4^ CFU/mL	53.69	11.46	17.59
	10^5^ CFU/mL	51.85	13.29	20.40
Condition 9 (MAM:AAM:NVP) (2:1:1)	Blank	52.48	-	-
	10 CFU/mL	40.14	12.34	23.51
	10^2^ CFU/mL	34.81	17.67	33.66
	10^3^ CFU/mL	31.60	20.88	39.78
	10^4^ CFU/mL	27.64	24.83	47.32
	10^5^ CFU/mL	24.58	27.89	53.15

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
