# Peer review of "Biosensors for Klebsiella pneumoniae with Molecularly Imprinted Polymer (MIP) Technique"

_sensors, 2022, doi:10.3390/s22124638_

Round 1
Reviewer 1 Report
The authors present the development of a MIP to modify an SPE electrode for the recognition of the bacterium Klebsiella pneumoniae.
The work could be interesting, but there are several points to clarify. Following the order of the manuscript:
1) in paragraph 2.1, how do they establish the exact number of cells present at the beginning of the serial dilutions?
2) In paragraph 2.3, it is not clear how it was established that 106 cells are required for the imprinting.
3) In paragraph 2.2, some doubts arise about how the GO in solution (which solvent?) can mix with the prepolymer that is already in the gel state, and why more GO is added directly to the electrode. Among other things, the fixation procedure of the bacterium onto the electrode surface should be reported here and not in paragraph 2.2.
4) The authors have tried numerous combinations of monomer ratios, but it would be interesting to know following which criterion they chose the combinations studied. Maybe a little experimental design would have helped.
5) Regarding the results of this experiment, it is not clear why the combinations n. 5 and 9 have been selected, since they themselves declare that such combinations "perform poorly for K. pneumoniae detection at a low concentration". Can they clarify better?
6) Why was CV chosen as a detection technique, and not a more performing pulsed technique? What is reported as the value of "I", the anodic, cathodic peak current or the area of the peaks?
7) Figure 8 shows the calibration lines on "gold electrodes" according to the caption of the figure and according to the values reported in table 5, but the slope of the red curve referred to combination “9” (4.096% relative current / log concentration) is different from that shown in figure 9 (7.51% relative current / log concentration) and also shown in the text.
8) An application to real samples is completely missing: in which matrix do the authors want to measure the presence of the bacterium? Since the work is dedicated to nosocomial infections one would think of an application in serum, but we do not know if such a sensor can work in this matrix….
Reviewer 2 Report
Review of a manuscript entitled “Nosocomial Infection Biosensors for Klebsiella pneumoniae with Molecular Imprinted Polymers (MIPs) Technique”
Nosocomial infection is one of the most important problems that occur in hospitals and reliable electrochemical sensors could be very helpful for quick detection. The authors of the study describe the development of an electrochemical sensor for the whole bacteria based on the screen-printed electrode.
The idea of the manuscript is interesting. The introduction is well written, but the methodology should be significantly improved. The parts of the results and discussions should be more organized.
Detailed comment for the authors:
The title of the manuscript arouses interest but it can be still improvable, especially because of the misspelling of the full name of MIP. It should be molecularly imprinted polymers.
Look through the manuscript to check for the correct name.
According to the principle, you obtained the polymer and imprints in it, I prefer to think that this technique more correctly should be named surface imprinted polymers SIP. Please look at these articles:
( http://dx.doi.org/10.1016/j.snb.2017.08.122 //
https://doi.org/10.1016/j.trac.2018.07.011 //
This one you have already cited: https://doi.org/10.3390/s20040996 )
English grammar should be polished.
In line 23 there is mentioned that the proposed sensor “can achieve a high linear response”. But you didn’t add the values of the linear range of this response. Please, include the values of linear response, and also, the LOD and the LOQ values.
The introduction and Discussion part could be advanced by a discussion of the most recent reviews/insights in the development of molecularly imprinted polymer-based sensors
(DOI: https://doi.org/10.3390/s22031282 //
DOI: https://doi.org/10.1039/D1AN00149C //
DOI: https://doi.org/10.3390/polym13060974 //
DOI: https://doi.org/10.3390/s17040708 //
DOI: https://doi.org/10.1149/2754-2726/ac612c ).
Please, check the Klebsiella pneumoniae in the text. It should be italic in the whole manuscript.
The introduction is well written and sounds good. Some more references could be included for the purpose to demonstrate the variety of applications of the MIP (medical, environmental, food industry, etc.).
Please check the numbers of bacteria cells in lines 149-150.
The data in table 1 are doubtfully useful for the discussion about pre-polymerization complex formation. It would be best if you thought to remove it at all or supplement it with some more useful characteristics.
Section 2 of the manuscript is dedicated to the experimental part. You should transfer any discussions from the experimental part to section 3, which is named results and discussion.
Line 168-172. Please remove any unnecessary information.
Line 188: what do you mean by the words “polymer condition”? Is it the composition of the monomer mixture?
It is not the usual procedure of MIP preparation. In order to obtain the MIP, the template, in your case bacteria, should be mixed with monomers and then start polymerization. Now you have a dry polymer (lines 194-195) on which you put on the bacteria. How can you prove that you get a MIP but not a layer of bacteria simply absorbed on top of the polymer? SEM images show the bacteria, but what about the imprints?
Table 2. the crosslinker has to be included in the table. You should add the ratio of total monomers and crosslinker. This parameter is very important for the rigidity parameters of the final structure of the polymer. Can you explain the logic of such amounts of monomers? Is it done randomly?
Line 215: please check the producer of AutoLab 215 PGSTAT302N.
The full names are used only they first appear in the text, in the following parts of the text are used abbreviations but not full names (Lines 244, 247, 212, 187-191, and in many more lines).
According to table 4, the linear response was obtained for the 8th polymer. For what reasons was it eliminated from the following experiments?
Please add the number of replicated measurements to figure 9?
Authors should include their own article in the list of references: https://doi.org/10.1016/j.matlet.2022.131973 which has a very similar design to this one.
Round 2
Reviewer 1 Report
The authors provided satisfactory answers to the reviewer's questions. However, I suggest removing "nosocomial infections" from the title as there is no real application of the biosensor so far.
Reviewer 2 Report
The authors made a significant improvement in the article manuscript.
But there are some minor corrections that should be made.
1. Could you add the DOI numbers to the references?
2. Please check the numeration of the figures. There is no Fig. 7.
3. A gap should be inserted between the last word in the sentence and the reference.
4. The usage of MIP and MIT terms is still a little bit complicated in the text. Please mind the terms: molecularly imprinted polymers and molecular imprinting technology.
5. The full terms should be used only they first time appear in the text. In the following text abbreviations should be used. Look for MIP and MIT in lines 71, 85, 89, and 95 and for LOD in lines, 28, 88, 96, 354, and 369.
6. Please specify how the ratio was calculated. I only guess that the ratio is of amount expressed in moles (n:n:n)? Am I right?
7. The figure you added to the cover letter is clearer and easier to understand than the figure you added to the manuscript. Can you change them?
8. Please rearrange the figures 8 and 9. All figures should be in the same style.
